# B-Helper Neutrophils in Regional Lymph Nodes Correlate with Improved Prognosis in Patients with Head and Neck Cancer

**DOI:** 10.3390/cancers13123092

**Published:** 2021-06-21

**Authors:** Ekaterina Pylaeva, Irem Ozel, Anthony Squire, Ilona Spyra, Charlotte Wallner, Magdalena Korek, Georg Korschunow, Maksim Domnich, Elena Siakaeva, Moritz Goetz, Agnes Bankfalvi, Stephan Lang, Benjamin Kansy, Jadwiga Jablonska

**Affiliations:** 1Department of Otorhinolaryngology, University Hospital Essen, University Duisburg-Essen, 45147 Essen, Germany; Ekaterina.Pylaeva@uk-essen.de (E.P.); irem.oezel@uk-essen.de (I.O.); ilona.spyra@uk-essen.de (I.S.); charlotte.wallner@gmx.de (C.W.); magda.korek@gmail.com (M.K.); georg.korschunow@gmx.ch (G.K.); maksim.domnich@uk-essen.de (M.D.); elena.siakaeva@uk-essen.de (E.S.); stephan.lang@uk-essen.de (S.L.); 2Institute of Experimental Immunology and Imaging, University Hospital Essen, University Duisburg-Essen, 45141 Essen, Germany; anthony.squire@uk-essen.de; 3Institute of Pathology, University Hospital Essen, University of Duisburg-Essen, 45147 Essen, Germany; Moritz.Goetz@uk-essen.de (M.G.); agnes.bankfalvi@uk-essen.de (A.B.); 4German Cancer Consortium (DKTK) Partner Site Düsseldorf/Essen, 45147 Essen, Germany

**Keywords:** neutrophils, B cells, B helper neutrophils, NBH, head-and-neck cancer, lymph nodes, regional lymph nodes, head-and-neck squamous cell carcinoma

## Abstract

**Simple Summary:**

Neutrophils exhibit multiple functions during cancer progression and are believed to regulate adaptive immune responses to cancer. In addition to their interactions with T cells in this context, these cells are also believed to interact with B cells. Neutrophils have been found in the marginal zone of the spleen, where they exhibit helper cell characteristics, supporting B cell proliferation and activation. Here, we investigate the effect of neutrophils on B cells in the regional lymph nodes (RLN) of head-and-neck cancer (HNC) patients. We have identified that, in RLNs, neutrophils express a helper cell phenotype that was associated with the increased activation and proliferation of B cells. Importantly, the high abundance of neutrophils in the B cell follicles of regional lymph nodes is associated with significantly improved HNC patient survival.

**Abstract:**

The role of neutrophils during cancer formation and elimination is diverse. Here, for the first time, we investigate neutrophil helper cells (N_BH_), their influence on B cell activity in the regional lymph nodes (RLN) of head-and-neck cancer patients and the effect of this neutrophil/B cell interaction on patient prognosis. Circulating and RLN neutrophils of patients with stage I–IV head-and-neck squamous cell carcinoma were investigated with flow cytometry and qPCR. In addition, neutrophil/B cell co-localization in RLNs was evaluated using immunohistochemistry. B cell proliferation was assessed and correlated with the distance to neutrophils. Patient survival was evaluated. Neutrophils with the helper cell phenotype were identified in the RLN of HNC patients. B cells in close proximity to such N_BH_ showed significantly higher proliferation rates, together with elevated activation-induced cytidine deaminase (AID) expression. Notably, patient survival was significantly higher in individuals with high N_BH_ frequencies in the B follicles of RLNs. Neutrophils in RLN can support T cell-independent activation of the adaptive immune system through B cell stimulation, capturing helper cell phenotype character. The presence of such helper neutrophils in the RLNs of HNC patients positively correlates with patient prognosis.

## 1. Introduction

Over the years, it has become clear that neutrophils are not solely short-lived, innate responder cells, solely responsible for acute innate immune responses to bacterial infections. The role of neutrophils as mediators between innate and adaptive immune responses has become known [1]. Therefore, the plasticity of neutrophils has been intensely investigated, leading to the identification of distinct neutrophil subsets in cancer [2], with tumor-supportive (pro-angiogenic, pro-metastatic, immunosuppressive) or anti-tumor properties (cancer-cell killing, immunostimulatory) [3,4,5].

The interaction of neutrophils with the adaptive immune system was shown in the cytokine-dependent activation of antigen-presenting cells with TNFα [6], as well as both the activation and inhibition of T cells in tumor tissue [7].

In the past decade, a T cell-independent mechanism of neutrophil-mediated regulation of the adaptive immune system has surfaced. In the splenic marginal zone (MZ), a subpopulation of neutrophils has been identified and classified as B cell-helper neutrophils (N_BH_) due to their capacity to stimulate B cells through the expression of proliferation-inducing ligand (APRIL) and B cell-activation factor (BAFF) [7].

In tumors, B cell-mediated adaptive immune responses can play a beneficial role [8]. The anti-cancer activity of B cells is achieved through the secretion of immunoglobulins, promotion of T cell responses and direct cancer-cell killing [9]. Additionally, B cells are involved in the formation of tertiary lymphoid structures (TLS) that are associated with improved prognosis in cancer patients [10].

So far, little is known about the influence of neutrophils on B cell activity in cancer patients. This led us to investigate neutrophil/B cell interactions in the regional lymph nodes (RLNs) of head and neck cancer (HNC) patients in order to unravel their role in cancer progression and patient prognosis.

## 2. Materials and Methods

Patients: The protocol was approved by the Ethics Committee of the University Duisburg-Essen. All subjects gave written informed consent in accordance with the Declaration of Helsinki.

A total of 71 patients with stage I–IV head and neck squamous cell carcinoma who were scheduled for surgical resection consented to the tissue collection of a portion of the lymph nodes, tumor and/or blood for research purposes at the Otorhinolaryngology Department of the University Hospital of Duisburg-Essen, Essen, Germany. Patients with unknown or other histological types of tumors like adenocarcinoma, sarcoma, lymphoma, cancer with unknown primary origin (CUP) and nasopharyngeal squamous cell carcinoma with a different standard of care, as well as patients who received immunosuppressive therapies prior to the study were not included. All fresh tissue samples were obtained after pathological examination for clinical reasons; all histological findings (the type of the malignancy and metastatic status of RLN) were proven by experienced pathologists.

We introduced the grouping of LNs based on the presence of metastasis in the given sample as well as in the other LNs of the patient: “met−−” (refers to N0 stage: no metastasis was detected either in the given LN or in the other evaluated LNs), “met+−” (N1–3 stage with proven nodal metastasis but not in the given LN) and “met++” (N1–3 stage with proven nodal metastasis in the given LN).

Detailed characteristics of the patients are presented in Table 1.

Human sample collection and storage: Fresh RLN tissue samples were stored in DMEMc (DMEM (Gibco, Life Technologies/Thermo Fisher Scientific, Karlsruhe, Germany) containing 10% FCS and 1% penicillin–streptomycin) on ice for a maximum of 30 min before processing. Human peripheral blood was drawn into 3.8% sodium citrate anticoagulant monovettes and stored at room temperature for a maximum of 2 h before processing.

Isolation of blood neutrophils and flow cytometry: Whole blood was separated by density gradient centrifugation (Biocoll density 1.077 g/mL, Biochrom, Berlin, Germany). The mononuclear cell fraction was discarded, and neutrophils (purity > 95%) were isolated by sedimentation over 1% polyvinyl alcohol, followed by hypotonic lysis (0.2% NaCl) of erythrocytes with the purity > 95%. The pellet was either incubated in a CD16/32 Fc block and further stained with the antibodies listed below or appropriate isotype controls or frozen in RNAlater (Ambion, Invitrogen, Hilden, Germany) for further RNA extraction and RT-qPCR.

Preparation of a single-cell suspension from RLNs and flow cytometry: Tissue samples (tRLNs) were digested using a dispase 0.2 µg/mL, collagenase A 0.2 µg/mL, DNase I 100 µg/mL (all Sigma-Aldrich/Merck, Taufkirchen, Germany) solution in DMEMc, 1 mL per sample. Cells were meshed through 100 µm sterile filters (Cell Trics, Partec, Norderstedt, Germany); the pellet was incubated in a CD16/32 Fc block and further stained with the antibodies listed below or appropriate isotype controls.

Antibodies used in the study: CD3 (HIT3a, BioLegend, Fell, Germany), CD19 (clone HIB19, BioLegend), CD66b (G10F5, BioLegend) and DAPI (BioLegend) antibodies were used for the detection of T cells, B cells, neutrophils and cell nuclei in lymph nodes, respectively. AID (EK2-5G9, BD Bioscience, Heidelberg, Germany), rat IgG2b (RMG2b-1, BioLegend) and Ki-67 (Ki-67, BioLegend) antibodies were used to assess the proliferation and activation of B cells in the lymph nodes, respectively. CD11b (M1/70, BioLegend), CD11c (Bu15, BioLegend), CD16 (3G8, BioLegend), CD54 (ICAM1) (HA58, BioLegend), CD62L (DREG-56, BioLegend), CD66b (G10F5, BioLegend), CD80 (2D10, BioLegend), CD86 (IT2.2, BioLegend), CD257 (BAFF, BLYS) (1D6, BioLegend), CXCR2 (5E8/CXCR2, Biolegend), MHCI (BB7.2, BioLegnd), MHCII (L243, BioLegend) and Viability Dye eFluor 780 (eBioscience) antibodies were used for the phenotyping of both blood and lymph-node neutrophils.

Isolation of tissue neutrophils: Neutrophils were isolated from RLN single-cell suspensions using flow sorting (FACS Aria cell sorter (BD Biosciences, BD)), based on the phenotype (single alive CD66b^+^cells) with the purity > 95%. Cells were frozen in RNAlater for further RNA extraction and RT-qPCR.

Cytospins: Cytospins of isolated CD66b+ RLN neutrophils on poly-D-lysine coated glass slides were prepared. The morphology of neutrophils was assessed after Giemsa staining.

RLN-conditioned medium and in vitro stimulation of blood neutrophils: met−−, met+− and met++ RLNs (n = 3 in each group) were cut and incubated in cell culture media (0.06 g of tissue in 0.2 mL DMEMc) for 4 h, RLN-conditioned medium was collected and stored at −80 °C until use. Isolated blood neutrophils from the healthy volunteers (*n* = 6) were incubated in control DMEMc or RLN-conditioned medium for 18 h at +37 °C and 5% CO_2_, then the viability and expression of BAFF and APRIL were estimated with flow cytometry; cells were frozen in RNAlater for further RNA extraction and RT-qPCR.

Quantitative RT-PCR: Samples (isolated blood and RLNs neutrophils, *n* = 12) were washed in PBS; the pellet was re-suspended in RNAlater and stored at −20 °C. RNA was isolated using RNeasy Mini Kit (Qiagen, Hilden, Germany) according to the manufacturer’s protocol. RT-qPCR was performed using primers listed in Table 2. The bactin housekeeping gene was used. mRNA expression was measured using the SYBR green qPCR kit and the absolute and relative gene expressions were calculated with 2^−ΔCt^ (for comparing blood versus RLN neutrophils) and 2^−ΔΔCt^ (for gene expressions in neutrophils that were treated with control DMEMc or RLN-conditioned medium) formulations.

Paraffin sections: For histopathological evaluation, paraffin blocks were cut using the Thermo Scientific Microm HM 340E, with a thickness of 4 µm. Samples were deparaffinized. A CD66b IgM mouse-anti-human antibody (BioLegend) was used to mark neutrophilic granulocytes with horseradish peroxidase functioning as the immunoenzyme for reacting with the AEC-substrate (Invitrogen) turning the cells red. The CD3 rabbit-anti-human-antibody was used to mark T-cells and secondarily, an alcalic phosphatasis reaction with the IHC green chromogen (BioVision, California, CA, USA) staining the cells green. Counterstaining of nuclei was achieved by hematoxylin. The samples were scanned via the Aperio ScanScope AT2 (Leica Biosystems, Wetzlar, Germany). The scans in the format ScanScope Virtual Slide (SVS) show a ×20 magnified view of the whole layer on the slide. Three to five random B follicles per sample were analyzed, the number of neutrophils per 0.01 mm^2^ was estimated. Patients were divided into high and low infiltration groups based on the median number. In the end, those high and low neutrophil counts were compared to the corresponding days of survival. The follow-up was performed though the cancer aftercare program in our clinic, which is based on frequent controls after tumor therapy. For the first two years, we have a patient follow-up every two months, with decreasing frequency until year five.

Cryosections: Human RLNs were embedded with Tissue-Tek O.C.T. Compound (Sakura Finetek, Staufen, Germany) and snap frozen at −80 °C. Cryosections measuring 4 μm were fixed with ice-cold acetone, stained with anti-CD66b, anti-CD19, anti-AID or anti-Ki67 secondary antibodies and DAPI, dried and mounted with Neo-Mount (Merck). Microscopy was performed using a Zeiss AxioObserver.Z1 Inverted Microscope with ApoTome Optical Sectioning equipped with filters for: DAPI, FITC, Alexa Fluor 488, GFP, DsRed and Cy3. The acquired images were preprocessed, segmented and analyzed with a combination of an ImageJ FIJI script (Rasband, W.S., ImageJ, U. S. National Institutes of Health, Bethesda, https://imagej.nih.gov/ij/, (accessed on 17 June 2021)), Tissue Studio (Definiens, version 4.4.2) and a Python script (Python version 3.6.4, https://www.python.org, (accessed on 17 June 2021)), respectively. In ImageJ, a background subtraction of the Ki67, CD19 and CD66b channels was performed for all images using rolling ball background subtraction with a radius of 64 microns. The resulting background subtracted and background images for each image channel were saved to disc as separate image layers in the form of 16 bit tiff files. These image layers were loaded into the Definiens software package Tissue Studio and cell segmentation was performed using segmentation of the DAPI nuclear stain and a subsequent nuclear region growth of 1 micron. For every segmented cell, the image name, cell (x, y) coordinates and average intensities for the AID/Ki67, CD19 and CD66b signals and their respective backgrounds were output to a csv file. In Tissue Studio, a subselection of all images was used to determine the ratio of the cell signal intensity relative to the local background for individual cells considered positive for the CD19 and CD66b labels. These ratios were subsequently used to select CD19- and CD66b-positive cells in a subsequent analysis of the Tissue Studio csv file. The csv file was processed and analyzed using a home-written Python script making use of the numpy (NumPy. Available online: https://numpy.org (accessed on 17 June 2021)) and pandas (pandas. Available online: https://pandas.pydata.org (accessed on 17 June 2021)) modules. In the script, both the average AID/Ki67 intensity and cell density of CD19 positive cells as a function of increasing radius around each of the CD66b-positive cells was determined. These were then normalized to the corresponding values for all image regions. Thus, both the AID/KI67 signal and cell density for CD19 cells could be correlated with their proximity to CD66b cells.

Cell Viability: The viability of neutrophils was determined with an Annexin V/7-aminoactinomycin (7AAD) apoptosis detection kit (BD Biosciences). Isolated blood neutrophils from healthy donors were incubated in a control DMEMc or RLN-conditioned medium for 18 h at +37 °C and 5% CO_2_, then the percentages of living cells (Annexin V^−^/7-AAD^−^) were detected with a flow cytometer according to the kit’s instructions.

Statistics: The normality of the distribution was checked with the Kolmogorov–Smirnov Test. Descriptive statistics (number, percentage of patients (n, %) and mean ± standard deviation (m ± SD) for normally distributed parameters or median (interquartile range) for non-normal distribution) were used for patients’ clinical presentations. Normally distributed samples were compared using ANOVA for multiple comparison and Student t-tests for two independent samples. For nonparametrically distributed samples, a Kruskal–Wallis ANOVA with the Bonferroni correction for multiple comparisons and a Mann–Whitney U-Test for two independent samples or Wilcoxon test for dependent samples were used. For the comparison of frequencies in groups, a chi-squared test (with Yates’s correction for small data) was used. Correlations were analyzed with a Spearman R test. Kaplan–Meier curves for the survival function were compared via a log-rank test. *p* < 0.05 was considered significant.

## 3. Results

### 3.1. Neutrophils in the RLNs of HNC Patients Possess N_BH_ Phenotype

As RLNs constitute a major checkpoint for tumors, we first assessed the phenotype of RLN neutrophils. For this, we isolated CD66b^+^ cells from RLNs and morphologically verified them as neutrophils due to their segmented nuclei (Figure 1A). We showed that CD66b^+^ neutrophils constitute 0.1–10% of single living cells in RLNs (Figure 1B) and differ in their phenotype when compared to blood PMCs. We observed an activated phenotype of neutrophils in RLNs (CD66b^high^, CD11b^high^, CD16^dim^), with the upregulation of the molecules responsible for antigen presentation and lymphocyte activation (HLA-DR^+^, CD86^+^, CD11c^+^, ICAM1^+^) and with a significant decrease of the adhesion molecule CD62L and chemokine receptor CXCR2 (Figure 1C). Importantly, the gene expression of molecules known to specifically support B cell maturation and survival—*BAFF* (Figure 1D) and *APRIL* (Figure 1E)—were strongly upregulated in such RLN neutrophils as compared to circulating neutrophils.

As the activity of neutrophils could be altered by the RLN environment, we have isolated blood neutrophils and incubated them with LN supernatant and assessed changes in gene regulation and phenotype. Moreover, to assess the effect of metastasis on their activity, we compared the effect of supernatants derived from non-metastatic and metastatic LNs. We introduced the grouping of LNs based on the presence of metastasis in the given sample, as well as in the other LNs of the patient: “met−−” (refers to N0 stage, no metastasis was detected either in the given LN or in the other evaluated LNs), “met−−” (N1–3 stage with proven nodal metastasis but not in the investigated LN) and “met++” (N1–3 stage with proven nodal metastasis in the investigated LN).

We demonstrated that LN-derived factors, especially in RLNs with histologically proven nodal metastasis, strongly improved the survival of neutrophils in culture (Figure 2A). This observation correlated positively with the upregulation of STAT3 in neutrophils incubated in vitro with LN supernatants (Figure 2B,C). Interestingly, blood neutrophils, characterized by the low expression of CXCR5, increased their gene expression under stimulation with the LN-conditioned medium (Figure 2D). This could be involved in their migration into B-rich zones of RLNs. Moreover, a trend to increase *BAFF*, *APRIL* and *IL21* gene expression in blood neutrophils after stimulation with LN-conditioned medium was observed (Figure 2E–H), suggesting more profound stimulation of neutrophils in tissues than in vitro.

### 3.2. RLN Neutrophils Induce AID Expression in Neighboring B Cells

To further investigate the impact of neutrophils on RLN B cells, we have assessed neutrophils directly in LN tissue that was freshly isolated from HNC patients. We have used Definiens Tissue Studio software to determine nuclei and cell distribution in tissue sections. Thus, we were able to measure the distances between the investigated cells according to cell coordinates (Figure 3A).

Neutrophils co-localize with B cells in RLNs (Appendix A). Activated B lymphocytes express activation-induced cytidine deaminase (AID), therefore, we evaluated the expression of AID in B cells and correlated it with the proximity of neutrophils as depicted in Figure 3B. We showed the highest expression of AID in B cells localized in close proximity (less than 10 µm) to neutrophils and a decreased signal at further distances (Figure 3C–E). Only a minor signal of isotype control for AID was detected (Appendix A). No significant difference between AID expression in B cells between non-metastatic and metastatic RLNs was observed (Appendix A).

These findings suggest the cell–cell-contact involvement of neutrophils in the activation of B cells in RLNs.

### 3.3. RLN Neutrophils Stimulate the Proliferation of B Cells through Cell-Cell Contact

We have evaluated the expression of Ki67 in B cells in relation to the distance from neutrophils. We observed that the high infiltration of LN tissue by neutrophils was associated with the increased proliferative activity of B cells (Figure 4A). Moreover, spatial analysis revealed the highest expression of Ki67 in B cells localized in close proximity (less than 10 µm) to neutrophils, with progressive decreases at further distances (Figure 4B–E). Interestingly, no significant difference between Ki67 expression in B cells between non-metastatic and metastatic RLNs was observed (Appendix A). A positive correlation between the expression of AID and Ki67 by B cells (Figure 4C) suggests a connection between the AID-related adjustment of immunoglobulins and signals relevant for B cell expansion in RLNs.

### 3.4. High Abundance of Neutrophils in the B Cell Zones of RLNs Are Associated with the Improved Overall Survival of HNC Patients

Considering the stimulating influence of LN-infiltrating N_BH_ on B cell activation and proliferation, we further addressed the clinical significance of the abundance of N_BH_ in RLNs in HNC patients. To assess if the nodal stage has an influence on neutrophil infiltration, we evaluated the infiltration of the B cell follicles in RLNs by CD66b^+^ neutrophils. Three to five B follicles per LN section were analyzed; the density of infiltrating neutrophils was assessed, and the mean value per sample calculated.

The CD66b^+^ neutrophils were clearly detectable in the B cell areas of RLNs (Figure 5A). Interestingly, the number of neutrophils in B cell areas did not differ between metastatic LN and non-metastatic LNs (Figure 5B), since in all patient groups, different abundance of neutrophils in B-rich zones could be observed.

We have divided patients into two groups according to the extent of the infiltration of B follicles by neutrophils—into low (no neutrophils in B follicles detected) and high infiltration (Figure 5D,E). Table 3 provides an overview of their clinicopathological features. The mean survival (follow up) in the Low group was 840 days versus 1319 days in the High group (*p* = 0.01). Patients who survived longer than five years were calculated as surviving 1862 days (five years).

To investigate the clinical relevance of the above findings, we performed a Kaplan–Meyer analysis of the patients’ survival, depending on the level of neutrophils in B follicles. Importantly, we observed the significantly improved survival of patients with high neutrophil infiltration of B cell areas (Figure 5C), independent of the patients’ nodal status (Appendix A). This points to a significant influence of N_BH_ on the patients’ prognosis.

Altogether, our data confirm an important anti-cancer role of neutrophils in regional lymph nodes. Neutrophils can exhibit a B cell helper phenotype and activate essential B cell functions. Clinically, these observations are supported by the fact that patients showing high levels of neutrophil/B cell interactions demonstrate significantly improved survival.

## 4. Discussion

B cells and plasma cells were suggested to possess potent anti-cancer activity in different types of cancer [11,12,13,14], including HNC [15,16,17,18]. Neutrophils in the spleen were proven to exert B cell supportive functions in steady state and inflammatory diseases [7], although the role of this phenomenon in cancer remains unclear. Here, we have demonstrated the presence of neutrophils with a B helper cell phenotype in the B zones of RLNs, their ability to stimulate the activation and proliferation of B cells and their positive influence on patients’ prognosis with HNC.

Others have shown that the longevity of neutrophils can be increased through cytokines released in inflammatory conditions [19]. Moreover, the observation of mature neutrophils proliferating outside of the bone marrow, leading to increased tissue persistence [20], has supported the image-transition of neutrophils: after initially being reduced to sole responder cells, their role as effector cells with distinct interactions with both the native and adaptive immune responses became more and more clear. Besides residing in the bone marrow, neutrophils build significant tissue resident reservoirs in lymphoid organs, liver and lung [21], where their functions are not yet completely understood. Puga and colleagues described a new phenotype of neutrophils in the marginal zone (MZ) of the spleen, comparable to helper cells, interacting with MZ-resident B cells [7]. Balázs and colleagues have demonstrated that these granulocytes, together with dendritic cells, are bloodborne and home into the marginal zone upon antigen stimulation [22]. Here, they provide a T cell-independent pathway of immune activation, enhancing antigen-specific B cell responses, influencing immunoglobulin production, somatic hypermutation and class switching [7]. In mice, Deniset and colleagues reported both resident and “homing” populations of neutrophils in the MZ of the spleen that exerted helper cell functions for B cells [23].

Here, we reported the activated phenotype of neutrophils in RLNs (CD66b^high^ CD11b^high^ CD1^dim^), together with the elevated expression of molecules involved in antigen presentation and the activation of lymphocytes (HLA-DR, CD86, ICAM1) and B cells (BAFF, APRIL). The expression of BAFF and APRIL is characteristic for the key signaling pathways in N_BH_ cells identified so far [7,24,25]. BAFF acts by binding to the B cell maturation antigen (BCMA) receptor, the transmembrane activator and calcium modulator and cyclophilin ligand interactor (TACI), as well as the B cell-activating Factor (BAFF) receptor [26]. Through these receptors, BAFF acts as a survival and maturation factor on B cells. The overproduction of BAFF is associated with autoimmune disorders [27]. Therefore, its essential role for B cell signaling and activity is acknowledged. APRIL is part of the TNF ligand superfamily, binding to BCMA and TACI. Patients with APRIL deficiency display severe dysfunctions in plasma-cell maintenance, as well as immunoglobulin production [28]. Neutrophils are known as major sources of APRIL secretion and can therefore act as important B cell stimulators [29]. Other mechanisms exerted by N_BH_, important for B cell activation, include stimulation via CD40L-CD40, interleukin-21 (IL-21) and CXCL12, as well as an increase in NET production [7].

While the differences between phenotypes of circulating and RLN neutrophils were prominent in our study, in vitro stimulation of isolated blood neutrophils with LN-conditioned medium did not lead to a significant change in their activation. Nevertheless, we observed the STAT3-dependent increase of neutrophil viability under such stimulation. This might be due to stimulation by cytokines, such as IL-6, IL-8 and IL-10 [30], and colony-stimulating factors, such as granulocyte-macrophage colony-stimulating factor (GM-CSF) [31]. The minor effect on neutrophil gene and protein expression in these settings can be explained by the short lifespan of isolated blood neutrophils in vitro, too brief for the stimulation of sufficient concentrations of cytokines that could be detected. Therefore, this cannot be compared with the levels of cytokines locally available in tissues. Additionally, cell–cell contact with surrounding stroma cells in vivo might be required for proper activation [32].

To overcome limitations due to the in vitro situation, we have analyzed B cell activation in LN tissues in situ and correlated it with the proximity of these cells to neutrophils. Indeed, we observed the increased activation and proliferation of B cells localized in close proximity to neutrophils. This underlines the importance of the higher local concentrations of factors released by neutrophils and the necessity of cell–cell contact, with the involvement of adhesion molecules, such as ICAM1, which is known to contribute to the activation of B cells by promoting their adhesion and synapse formation between B cell receptors and antigens [33]. The importance of the interaction between B cells and neutrophils might be appraised by the investigation of Gätjen and colleagues, wherein they observed that the B cells of patients with chronic lymphatic leukemia skewed the neutrophils in the MZ of the lymph nodes into an N_BH_ phenotype in order to stimulate and promote proliferation of B cells [34]. Another evidence for the influence of N_BH_ on B cells is provided by the fact that patients with congenital neutropenia show significantly reduced levels of T cell-independent serum immunoglobulins, suggesting that the missing neutrophil stimulation of B cells results in significant functional (B cell) deficits [7].

So far, little is known about the B helper functions of neutrophils in a cancer setting. We were able to observe increased AID expression levels in B cells localized in close proximity to N_BH_ in regional lymph nodes. Of note, similar neutrophil frequencies in RLNs between metastatic and non-metastatic nodes were observed. This argues against nodal tumor-associated infiltration and strengthens the hypothesis of both residual N_BH_ cells and possibly “homed” N_BH_ cells upon antigen stimulation, as observed by Balázs and colleagues in an infectious antigen stimulation setting [22]. We suggest that the observed changes in the phenotypes of neutrophils in RLNs are based on the nodal tissue microenvironment. This microenvironment consists of a wide variety of cytokines and growth factors that have demonstrated their influence on neutrophil survival and differentiation [35].

Most intriguingly, cancer patients with low numbers of N_BH_ cells in RLNs featured a significantly worse outcome in HNC patients, with decreased survival rates. To our knowledge, we are the first to describe the association of the abundance of N_BH_ cells in RLN with cancer patients’ prognosis.

Neutrophils were demonstrated to possess both pro- and anti-cancer properties, depending on the tissue microenvironment. In gastric cancer, a negative prognostic significance for neutrophils infiltrating metastatic LNs was observed [36]. In bladder cancer, neutrophils were associated with the formation of a premetastatic niche in tumor-free lymph nodes [37], and neutrophil infiltration was reported to be STAT3 mediated.

Here, we observe the association of N_BH_ viability in RLNs with STAT3 upregulation in these cells. Several authors have described the pro-tumoral characteristics of STAT3 depending signaling [37,38], whereas others have observed the B cell activating properties of STAT3 signaling [7]. Possibly, these heterogeneous effects of STAT3 signaling can be attributed to post-translational modifications or cell metabolic conditions in the specific environment [39]. Interestingly, AID intensity and Ki67 intensity did not differ between metastatic and non-metastatic lymph nodes, and CXCR5, BAFF and APRIL expression in vitro did not correlate with STAT3, indicating tumor-cell-independent STAT3 signaling in the lymph nodes.

Hence, we observed—for the first time—neutrophils with a helper cell phenotype in regional lymph nodes of cancer patients, executing a stimulatory effect on CD19^+^ B cells and thus significantly improving their prognosis.

## 5. Conclusions

The role of the neutrophils in cancer formation and progression is diverse, strongly depending on the surrounding influence of the TME and the periphery. In the presented manuscript, we bring new insights on the beneficial role of a specific subset of neutrophil helper cells in tumors in the RLNs of HNC patients. Neutrophils in RLNs display a N_BH_ phenotype and stimulate B cells through the induction of their AID expression. This potentially supports the selection of B cells with the highest affinity against cancer antigens, resulting in a T cell-independent, yet antigen-specific, antibody response. Therefore, an abundance of B cell-supportive N_BH_ in RLNs suggests a beneficial prognosis in HNC. These observations provide a basis for immunotherapeutic approaches incorporating N_BH_/B cell interactions for T cell-independent tumor antigen-specific therapy.

## Figures and Tables

**Figure 1 cancers-13-03092-f001:**
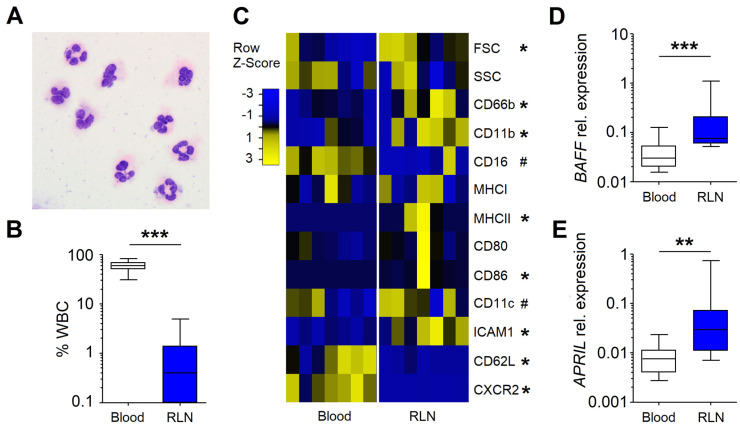
CD66b⁺ neutrophils infiltrate the RLNs of HNC patients and exhibit N_BH_ properties. (**A**) Giemsa staining of CD66b^+^ cells isolated from the RLNs of HNC patients shows their segmented nuclei. Scale bar = 10 µm. (**B**) The percentage of neutrophils from WBC, both in blood (white) and in RLNs (blue) from HNC patients (*n* = 7). (**C**) Heatmap of the cell phenotypes of blood and RLN neutrophils isolated from HNC patients, assessed with flow cytometry (*n* = 7). (**D**) Elevated *BAFF* gene expression in RLN neutrophils (blue) compared to blood neutrophils (white). (**E**) Elevated *APRIL* gene expression in RLN neutrophils (blue). A Mann–Whitney test for two independent samples and a Wilcoxon test for two dependent samples were used. Data are shown as median, 25–75 percentiles, min–max, *** *p* < 0.001, ** *p* < 0.01, * *p* < 0.05, # *p* = 0.06.

**Figure 2 cancers-13-03092-f002:**
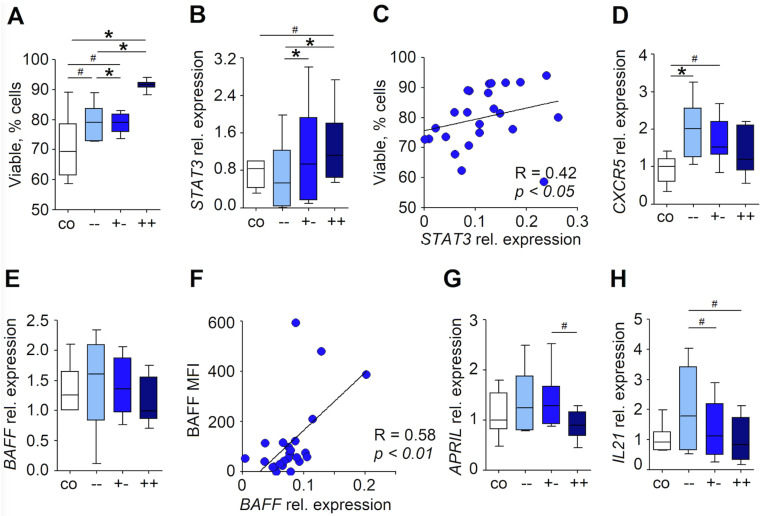
The RLN tissue microenvironment supports the N_BH_ properties of infiltrating neutrophils. Neutrophils isolated from healthy donors (*n* = 6) were cultured with cell-culture media and LN supernatants. DMEMc (co, white), non-metastatic LN supernatant (−−, light blue and +−, blue) and metastatic LN supernatant (++, dark blue). (**A**) Survival of neutrophils that are cultured in DMEMc (co, white), non-metastatic LN supernatant (−−, light blue and +−, blue) and metastatic LN supernatant (++, dark blue). (**B**) *STAT3* gene expression, relative to the *BACTIN* housekeeping gene, in neutrophils. Cells were cultured with cell-culture media and LN supernatants. DMEMc (co, white), non-metastatic LN supernatant (−−, light blue and +−, blue) and metastatic LN supernatant (++, dark blue). (**C**) Correlation between neutrophil survival and *STAT3* expression relative to *BACTIN*. (**D**) *CXCR5* gene expression, relative to the *BACTIN* housekeeping gene, in neutrophils. Wilcoxon test for dependent samples. (**E**) *BAFF* gene expression, relative to *BACTIN* housekeeping gene, in neutrophils. Cells were cultured with cell-culture media and LN supernatants. DMEMc (co, white), non-metastatic LN supernatant (−−, cyan and +−, blue) and metastatic LN supernatant (++, dark blue). (**F**) The correlation between the surface expression of the BAFF protein and *BAFF* gene expression relative to *BACTIN*. (**G**) *APRIL* gene expression, relative to the *BACTIN* housekeeping gene, in neutrophils. Cells were cultured with cell-culture media and LN supernatants. DMEMc (co, white), non-metastatic LN supernatant (−−, cyan and +−, blue) and metastatic LN supernatant (++, dark blue). (**H**) *IL21* gene expression, relative to the *BACTIN* housekeeping gene, in neutrophils. Cells were cultured with cell-culture media and LN supernatants. DMEMc (co, white), non-metastatic LN supernatant (−−, cyan and +−, blue) and metastatic LN supernatant (++, dark blue). Correlations were analyzed with a Spearman R test. Data are shown as median, 25–75 percentiles, min-max, * *p* < 0.05, # *p* = 0.06.

**Figure 3 cancers-13-03092-f003:**
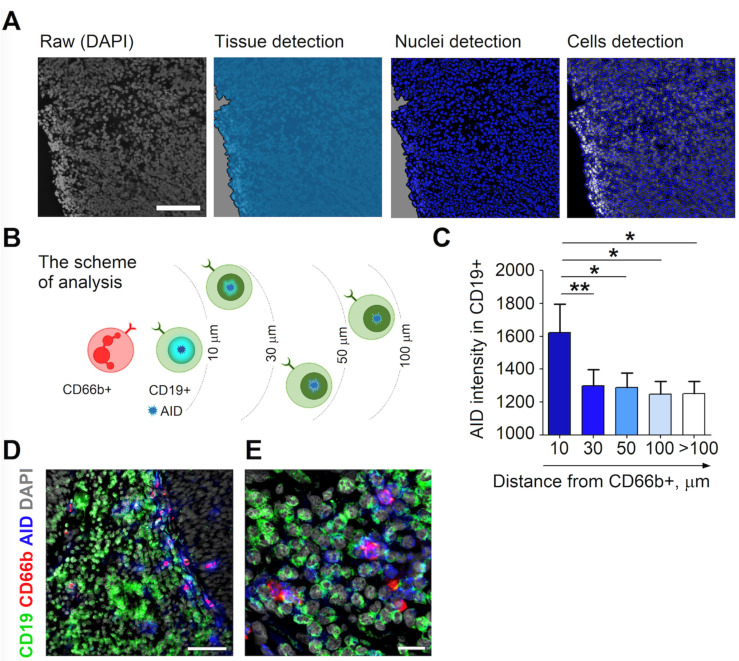
B cells in close proximity to N_BH_ express significantly higher levels of AID in the RLN of HNC patients. RLNs (*n* = 16) of HNC patients were frozen, cryosections stained, and the expression of AID was estimated in CD19^+^ cells. This was correlated with the distance from CD66b^+^ neutrophils. Images (numbering 3–5 per individual LN) were evaluated. (**A**) Representative images of tissue detection (light blue filling), nucleus detection (dark blue filling) and cell detection (dark blue line) in RLNs with Tissue Studio software. Scale bar 50 µm. (**B**) Schematic diagram of the scientific approach and analysis of the data from Tissue Studio software (**C**) Increased AID intensity in B cells with neutrophil vicinity. Neutrophil distance by 10, 30, 50, 100 and >100 µm are shown. (**D**,**E**) Representative images of the immunofluorescence staining of LN tissues for neutrophils (red), B cells (green) AID (blue) and DAPI (gray). Scale bar, 100 µm (**D**) and 20 µm (**E**). For nonparametrically distributed samples, a Kruskal–Wallis ANOVA with the Bonferroni correction for multiple comparisons and a Wilcoxon test for dependent samples were used. Data are shown as individual values and mean. ** *p* < 0.01, * *p* < 0.05.

**Figure 4 cancers-13-03092-f004:**
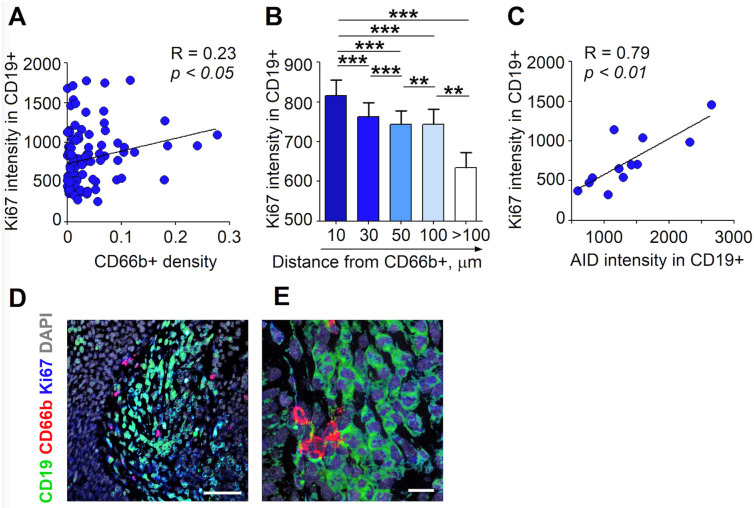
B cells in close proximity to N_BH_ express elevated levels of Ki67. RLNs (*n* = 16) of patients with HNC were frozen, and the expression of Ki67 was estimated in CD19+ cells in relation to the distance from CD66b^+^ on cryosections; 3–5 images per individual LN were evaluated. (**A**) Correlation between Ki67 expression in B cells with increasing neutrophil numbers. (**B**) High Ki67 intensity in B cells that are in close proximity to neutrophils. The intensity of Ki67 in B cells in the neutrophil vicinity of 10, 30, 50, 100 and >100 µm were shown. (**C**) Correlation between Ki67 and AID expressions in B cells. (**D**,**E**) Representative images of the immunofluorescence staining of LN tissues for neutrophils (red), B cells (green) Ki67 (blue) and DAPI (gray). Scale bar, 100 µm (**D**) and 20 µm (**E**). For nonparametrically distributed samples, a Kruskal–Wallis ANOVA with the Bonferroni correction for multiple comparisons and a Wilcoxon test for dependent samples were used. Data are shown as individual values and mean; correlations were estimated with a Spearman rank R test. *** *p* < 0.001, ** *p* < 0.01.

**Figure 5 cancers-13-03092-f005:**
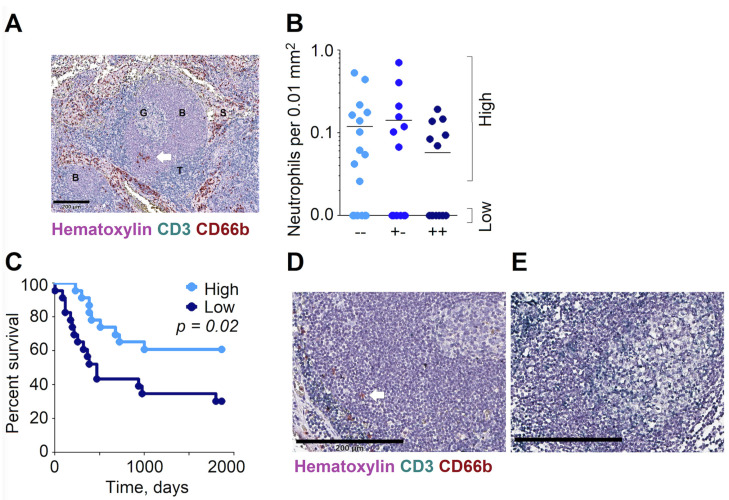
The high abundance of neutrophils in B cell follicles of RLNs significantly improves the overall survival of HNC patients. RLNs (*n* = 47) of patients with HNC were embedded in paraffin; the density of CD66b^+^ cells in B cell follicles was quantified in 3–5 follicles per image and correlated with the 5-year survival rate of the patients. (**A**) Immunohistochemistry of human RLNs. B: B zones, S: sinuses, T: T zones, G: germinal center. Red: CD66b+ neutrophils, violet/purple: B cells, green: CD3+ T cells. Exemplified figure in 200 µm scale bar. (**B**) Neutrophil counts in non-metastatic (“met− −” (cyan) and “met+−” (light blue)) and metastatic LNs (“met++” (dark blue)). (**C**) Overall survival of HNC patients’ is increased in high (light blue) neutrophil count when compared to low (dark blue) neutrophil count. (**D**,**E**) Immunohistochemistry of human RLNs. Exemplified images of the high and low infiltration of B cell follicles by neutrophils. Red: CD66b+ neutrophils, violet/purple: B cells, green: CD3+ T cells. Scale bar, 200 µm (**D**) and 100 µm (**E**). Kaplan–Meier curves for the survival function were compared via a log-rank test.

**Table 1 cancers-13-03092-t001:** Clinical characteristics of the patients enrolled.

	Cohort 1	Cohort 2
	(Flow Cytometry, Cryosections)	(Paraffin Sections)
	*n* = 16	*n* = 43
Age, years	62 (45–79)	61 (55–67)
Gender, male (*n*, %)	12 (75%)	31 (72%)
Analyzed LNs (*n*)	16	47
Localization (*n*, %):		
-hypopharynx	0	8 (19%)
-larynx	4 (25%)	9 (21%)
-nose	2 (12.5%)	0
-oral cavity	4 (25%)	0
-oropharynx	6 (37.5%)	24 (55%)
-pharynx	0	2 (5%)
T stage (*n*, %):		
−1	6 (37.5%)	14 (33%)
−2	6 (37.5%)	12 (28%)
−3	2 (12.5%)	10 (23%)
−4	2 (12.5%)	7 (16%)
n stage (*n*, %):		
0	6 (37.5%)	19 (44%)
−1	4 (25%)	5 (12%)
−2	5 (31.25%)	18 (42%)
−3	1 (6.25%)	1 (2%)
HPV-positive (*n*, %)	*n*/A	*n*/A

**Table 2 cancers-13-03092-t002:** Overview of used primers for qPCR.

Gene	Forward Primer	Reverse Primer
*APRIL*	5′-CGGAAAAGGAGAGCAGTGCTCA-3′	5′-GCCTAAGAGCTGGTTGCCACAT-3′
*BACTIN*	5′-AGCGGGAAATCGTGCGTG-3′	5′-GGGTACATGGTGGTGCCG-3′
*BAFF*	5′-ACCACGCGGAGAAGCTGCCAG-3′	5′-CTGCTGTTCTGACTGGAGTTGC-3′
*CXCR5*	5′-TGAAGTTCCGCAGTGACCTGTC-3′	5′-GAGGTGGCATTCTCTGACTCAG-3′
*IL21*	5′-CCAAGGTCAAGATCGCCACATG-3′	5′-TGGAGCTGGCAGAAATTCAGGG-3′
*STAT3*	5′-CTTTGAGACCGAGGTGTATCACC-3′	5′-GGTCAGCATGTTGTACCACAGG-3′

**Table 3 cancers-13-03092-t003:** Overview of the clinicopathological features of patients with high and low neutrophil infiltrations in the B cell zones of RLN.

	High	Low	*p*
Mean age (years)	61.8	61.4	0.98
Gender, male (%)	70	75	0.92
UICC Stage:			
UICC Stage I (%)	22	8	0.2
UICC Stage II (%)	13	8	0.6
UICC Stage III (%)	0	8	0.16
UICC Stage IV (%)	65	76	0.76
Grade:			
G2 (%)	71	87.5	0.18
G3 (%)	29	12.5	0.18
Smoking (%)	77	61	0.4
Alcohol (%)	31	54	0.23
Localization:			
Oropharynx (%)	79	42	0.02
Hypopharynx (%)	13	25	0.3
Larynx (%)	4	29	0.06
Nasopharynx (%)	4	4	0.97
Adjuvant therapy:			
RCT (%)	31	42	0.42
CT (%)	17	0	0.1
RT (%)	13	21	0.47
No (%)	39	37	0.93

## Data Availability

The data presented in this study are available on request from the corresponding author. The data are not publicly available due to restrictions (privacy).

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
