# Peer review of "B-Helper Neutrophils in Regional Lymph Nodes Correlate with Improved Prognosis in Patients with Head and Neck Cancer"

_cancers, 2021, doi:10.3390/cancers13123092_

Round 1
Reviewer 1 Report
Knowledge of cancer immunology is of the utmost importance. Only in the beginning of this millennium we started to understand in more detail the immunological microenvironment of cancer and still we have a lot to learn. In this study the authors have nicely shown the significance of neutrophils in the activation process.
The authors have done nice work at the laboratory, but there is a problem with the design of the study. It is known that SCCHN spread primarily to cervical lymph nodes and usually the areas of spreading are predictable but not always. Therefore a sentinel lymph node technique is sometimes used to detect the draining lymph node(s). In this study the authors don’t describe that any tracer or other techniques would have been used to detect the draining lymph node. In fact not even the neck dissection level of the collected lymph node (in comparison to primary site) is reported. Therefore it remains unclear if the positive survival effect of neutrophil infiltration in the lymph node is specific and local or is it a sign of regional or even systemic activation of the immune system. The comparison to blood neutrophils does not give the answer as the expression profile of almost all immune cells is different when they are circulating. Therefore it would be more appropriate to talk about regional lymph nodes than draining lymph nodes.
One minor point: It is illogical to exclude nasopharyngeal carcinomas due to EBV etiology and still include HPV positive oropharyngeal carcinomas.
Reviewer 2 Report
This manuscript aims to investigate the effect of neutrophils on B cells in tumor-draining lymph nodes (TDLN) of head-and-neck cancer (HNC) patients in order to define its potential significance. The topic is relevant since ponts to better define the role of a subclass of neutrophils in adaptive immune response. The manuscript is well written although material and methods lack of a clear description and in some experiments it is not clear how many patients have been analyzed. For sure this part needs to be improved. In addition several conclusions are not supported by the evidence provided.
Minor comments:
- Authors always says recently when refer to reference 7. This work has been published in 2011 and I would not say recently.
- Line 100: min should be read minutes (min). Also later hours is sometimes reported as h and sometimes as hours.
- Line 101 3,8 should be read 3.8. Same comment applies to other part of the manuscript.
- Antibodies used in the study: it is not clear how several of the antibodies reported are used to.
- were calculated by 2^-ΔCt and 149 2^-ΔΔCt formulations should be better define.
- Authors should decide in material and methods whether to report or not city and country of the Companies were reagents are made.
Major comments:
- Sentences: TDLNs constitute the major check point for tumors, filtrating the circulating tumor 214 cells and tumor antigens, and hosting anti-cancer adaptive T and B cell responses [11]. 215 Neutrophils were shown to infiltrate LNs in steady state [12] as well as in cancer [13], 216 although their effect on tumor progression and lymph node metastasis remains unclear. This part should be not included in result section.
- Figure 1 C: it is not clear how Heatmap is obtained. Is that obtained by flow cytometry analysis or gene expression? How many patients were analyzed? Same comment for Figure 1B
- Figure 2 A: how cell viability has been assessed?
- Blood neutrophils migrating to TDLNs gain the B helper cell phenotype, characterized with the potential to stimulate B cells. The conclusion is not supported by the evidence provided. First, activation of STAT3 and increased expression of CXCR5 do not diplay similar trend in --, +- and ++ LN. Second BAFF, APRIL and IL21 gene expression display just a trend in increasing and should be provide not as a supplmentary file. Swiching of neutrophils to NHB by activation of STAT3 signaling requires evaluation of IL-10 level as well as levels of BAFF, APRIL and IL21 in the LN supernatants and medium culturing of neutrophils following LN supernatant incubation, respectively.
- Line 281 -286: This part should be not included in result section.
- These findings suggest the cell-cell contact involvement of neutrophils in the activation of B cells in TDLNs. The conclusion is not supported by the evidence provided. Single staining and overalap staining should be shown. Some red dots do not appear to be associated with green and blue dots. Second no significant difference between AID expression in B cells between non-metastatic and metastatic TDLNs is observed. Third no isotype control for AID is provided. Again APRIL, IL21 and BAFF staining should be also provided since switching of B cell function is reported to be also mediated by indirect contact.
- Line 314 -318: This part should be not included in result section.
- Figure 4. The quality of images is very poor. No significant difference between Ki67 expression in B cells between non-metastatic and metastatic TDLNs is observed.
- This confirms the essential role of NBH in the suppression of cancer progression. The conclusion is not supported by the evidence provided and represents one of the major issue to the present work. Association of NBH cells with patient prognosis is too speculative. The authors have just analyzed a small number of patients which display different clinico-pathological characteristics such as Tumor stage and anatomical site. Even more no data related to follow up of patients included are provided as well as treatment modalities which may affect patient survival. Population is so etherogenous. The univariate analysis is not appropriate in order to define the potential title “B-helper Neutrophils in Tumor-draining Lymph Nodes Improve Prognosis in Patients with Head and Neck Cancer” . A multivariate analysis and more accurate presentation of clinico-pathological characteristics of the patients included are needed.
Round 2
Reviewer 2 Report
The manuscript has been revised and improved. However the conclusion "this confirms the essential role of NBH in the suppression of cancer progression" still need more data. The authors included some of clinico pathological data of patient population analyzed for prognosis. However no data related to follow up as well as events are reported. In addition the two groups of patiens are unbalaced. I would suggest to report follow up and patient outcomes. Second to analyze patient survival by including only stage IV patients and excluding stage I-II-III. .
Round 3
Reviewer 2 Report
1) Table 3: sum of tumor localization for the high group is 90..... it should be 100. The table needs to be revised. Table 3 should be described in the result section along with Kaplan Meier results and not in the methods.
2) No data related to follow up have been provided. Please provide the median follow up with range of patients.
Author Response
1) Table 3: sum of tumor localization for the high group is 90..... it should be 100. The table needs to be revised. Table 3 should be described in the result section along with Kaplan Meier results and not in the methods.
We thank the reviewer for the notification, the table was revised. We moved the table toe the results section, as suggested
2) No data related to follow up have been provided. Please provide the median follow up with range of patients.
Follow up was 5 years or until time of death. A more detailed description was added to the methods section: The mean survival (follow up) in the Low group was 840 days versus 1319 days in the High group (p=0.01), patients that survived longer than five years were calculated as 1862 days (five years) follow up.